# A novel function of R-spondin1 in regulating estrogen receptor expression independent of Wnt/β-catenin signaling

Ajun Geng[1†], Ting Wu[1†], Cheguo Cai[2†], Wenqian Song[1], Jiqiu Wang[3], Qing Cissy Yu[1*], Yi Arial Zeng[1,4*]

[1]State Key Laboratory of Cell Biology, CAS Center for Excellence in Molecular Cell Science, Institute of Biochemistry and Cell Biology, University of Chinese Academy of Sciences, Shanghai, China; [2]Medical Research Institute, Wuhan University, Wuhan, China; [3]Department of Endocrinology and Metabolism, Ruijin Hospital, Shanghai Jiao Tong University School of Medicine (SJTUSM), Shanghai, China; [4]School of Life Science, Hangzhou Institute for Advanced Study, University of Chinese Academy of Sciences, Chinese Academy of Sciences, Hangzhou, China

**Abstract** R-spondin1 (Rspo1) has been featured as a Wnt agonist, serving as a potent niche factor for stem cells in many tissues. Here we unveil a novel role of Rspo1 in promoting *estrogen receptor alpha (Esr1)* expression, hence regulating the output of steroid hormone signaling in the mouse mammary gland. This action of Rspo1 relies on the receptor Lgr4 and intracellular cAMP-PKA signaling, yet is independent of Wnt/β-catenin signaling. These mechanisms were reinforced by genetic evidence. Luminal cells-specific knockout of *Rspo1* results in decreased *Esr1* expression and reduced mammary side branches. In contrast, luminal cells-specific knockout of *Wnt4*, while attenuating basal cell Wnt/β-catenin signaling activities, enhances *Esr1* expression. Our data reveal a novel Wnt-independent role of Rspo1, in which Rspo1 acts as a bona fide GPCR activator eliciting intracellular cAMP signaling. The identification of Rspo1-ERα signaling axis may have a broad implication in estrogen-associated diseases.

**\*For correspondence:**
cissyyu@sibcb.ac.cn (QCY);
yzeng@sibcb.ac.cn (YAZ)

[†]These authors contributed equally to this work

**Competing interests:** The authors declare that no competing interests exist.

## Introduction

Estrogen and progesterone are the main players in mammary development and the progression of breast cancers (*Hilton et al., 2018*; *Macias and Hinck, 2012*). Both hormones act through their cognate receptors, estrogen receptor (ER) and progesterone receptor (PR) (*Hilton et al., 2018*). The mechanisms of ERα activity have been extensively studied (*Carroll, 2016*). However, the upstream regulation of ERα (*Esr1*) expression is much less understood.

The mammary gland is an epithelial organ profoundly influenced by estrogen and progesterone. The mammary gland is composed of basal and luminal cells, which can be separated by surface expression of CD24 and CD29/CD49f (*Shackleton et al., 2006*; *Stingl et al., 2006*). ER⁺ or PR⁺ cells, consisting 30 ~ 50% of luminal cells, can be enriched by surface expression of Sca1 (*Regan et al., 2012*; *Shehata et al., 2012*; *Sleeman et al., 2007*). Hormones exert their mitogenic effects primarily through induction of local growth factors (*Asselin-Labat et al., 2010*; *Brisken et al., 2000*; *Cai et al., 2014*; *Joshi et al., 2010*; *Rajaram et al., 2015*).

R-spondin1 (Rspo1) has been identified as a hormone-mediated local factor, whose expression is upregulated by estrogen and progesterone (*Cai et al., 2020*; *Cai et al., 2014*). R-spondin protein family (Rspo1-4) have been reported to function as niche factors for adult stem cells in multiple organs (*Greicius et al., 2018*; *Han et al., 2014*; *Planas-Paz et al., 2016*; *Sigal et al., 2017*), and Rspo1 has been implicated as critical growth factor in many in vitro stem cell expansion systems,

including intestine, stomach and liver (*Barker et al., 2010*; *Huch et al., 2013*; *Kim et al., 2005*; *Sato et al., 2009*). The role of Rspo1 in Wnt signaling has been extensively studied. Rspo1, through its interaction with its receptors Lgr4/5/6, enhances Wnt signaling by attenuating the turnover of Wnt receptors (*Hao et al., 2012*; *Koo et al., 2012*) and potentiating phosphorylation of the Wnt co-receptor Lrp (*Carmon et al., 2011*; *de Lau et al., 2011*; *Glinka et al., 2011*; *Gong et al., 2012*). In the mammary gland, Rspo1 synergizes with another niche factor, Wnt4, to promote mammary basal stem cell self-renewal (*Cai et al., 2014*). In line with the role of Rspo1 in MaSC regulation, Rspo1 expression is enhanced in the diestrus phase of the estrous cycle and during pregnancy (*Cai et al., 2014*), coinciding with the rise of progesterone level and the expansion of basal stem cells (*Asselin-Labat et al., 2010*; *Joshi et al., 2010*). Our recent study also reported the enhanced Rspo1 expression in estrus, a stage with high estrogen signaling activity (*Cai et al., 2020*). Another role of Rspo1 may exist besides maintaining basal stem cells.

In this study, we uncover a novel function of Rspo1 distinct from its previously reported role in stem cell regulation. We provide evidence that Rspo1 promotes ERα (*Esr1*) expression in luminal cells of the mammary gland. This action of Rspo1 is through activating G-protein coupled cAMP/ PKA pathway, while independent of Wnt/β-catenin signaling. Our data reveal a novel Wnt-independent role of Rspo1, and a new upstream regulatory axis for *Esr1* expression.

## Results

### Rspo1 induces ERα expression and promotes ERα signaling

To investigate the potential role of Rspo1 in luminal cells, we isolated primary luminal cells (Lin⁻, CD24⁺, CD29$^{lo}$) by FACS (fluorescence-activated cell sorting), and cultured them in 3D Matrigel in the presence of RSPO1 (0.5 µg/ml) (*Figure 1—figure supplement 1a*). Transcriptome and Gene ontology (GO) analysis identified enrichment of various features, including estrogen receptor activity (*Figure 1a and b*). qPCR analysis verified that the expression of ERα signaling target genes, including *Pgr* (progesterone receptor, PR), *Ctsd1* (Cathepsin D1) (*Meneses-Morales et al., 2014*), *and Wisp2* (*Zhang et al., 2012b*) are enhanced in the presence of RSPO1 (*Figure 1—figure supplement 1b*).

To further investigate how Rspo1 regulates ERα signaling, we isolated ER⁺ luminal cells (Lin⁻, CD24⁺, CD29$^{lo}$, Sca1⁺) and ER⁻ luminal cells (Lin⁻, CD24⁺, CD29$^{lo}$, Sca1⁻) based on Sca1 expression (*Figure 1c*), and cultured them in 3D. RSPO1 treatment resulted in the upregulation of ERα targets, *Pgr*, *Ctsd1* and *Wisp2* in ER⁺ luminal cells, indicating the further activation of ERα signaling (*Figure 1d*). Interestingly, the expression of ERα itself (*Esr1*) is also enhanced (*Figure 1e*). In contrast, ER⁻ luminal cells did not respond to RSPO1 stimulation (*Figure 1—figure supplement 1c*). Estrogen (Estradiol-E2, E2) is one of the few known upstream regulator of *Esr1* (*Chu et al., 2007*; *Kanaya et al., 2019*). Thus, E2 (1 µM) was used as control to show the extent of *Esr1* activation. We found that in this ER⁺ luminal cell culture system, RSPO1 elevated the expression of *Esr1* and its target *Pgr* to a level comparable with E2 treatment (compare *Figure 1d–e* with *Figure 1f*). The upregulation of ERα protein by RSPO1 was confirmed by Western blot analysis (*Figure 1g*). This role of RSPO1 was further validated in mouse mammary Eph4 cells. RSPO1 upregulates the expression of *Esr1* and ERα signaling targets *Pgr* and *Greb1* (growth regulation by estrogen in breast cancer 1) in a dose-depending manner (*Figure 1—figure supplement 2a–c*).

To investigate whether Rspo1 regulates *Esr1* transcription, we utilized a luciferase reporter driven by the proximal promoter (promoter A) of human *ESR1* (*Tanimoto et al., 1999*). We found that RSPO1 can induce luciferase expression in a dose-dependent manner, while the control reporter lacking *ESR1* promoter was not activated in any conditions (*Figure 1h*). Together, these data suggest that Rspo1 enhances *Esr1* transcription.

### Rspo1-induced ERα expression is dependent on Lgr4

To investigate the mechanisms through which Rspo1 regulates *Esr1*, we first examined which receptor of Rspo1 is involved. qPCR analysis indicated that all three Lgr receptors, Lgr4/5/6 are expressed in basal cells, but only Lgr4 is expressed in luminal cells (*Figure 2a*), suggesting that Rspo1 may rely on Lgr4 to signal in luminal cells in the context of *Esr1* induction. Within the luminal compartment, Lgr4 was evenly distributed in ER⁺ (Sca1⁺) and ER⁻ (Sca1⁻) luminal cells (*Figure 2a*). In situ

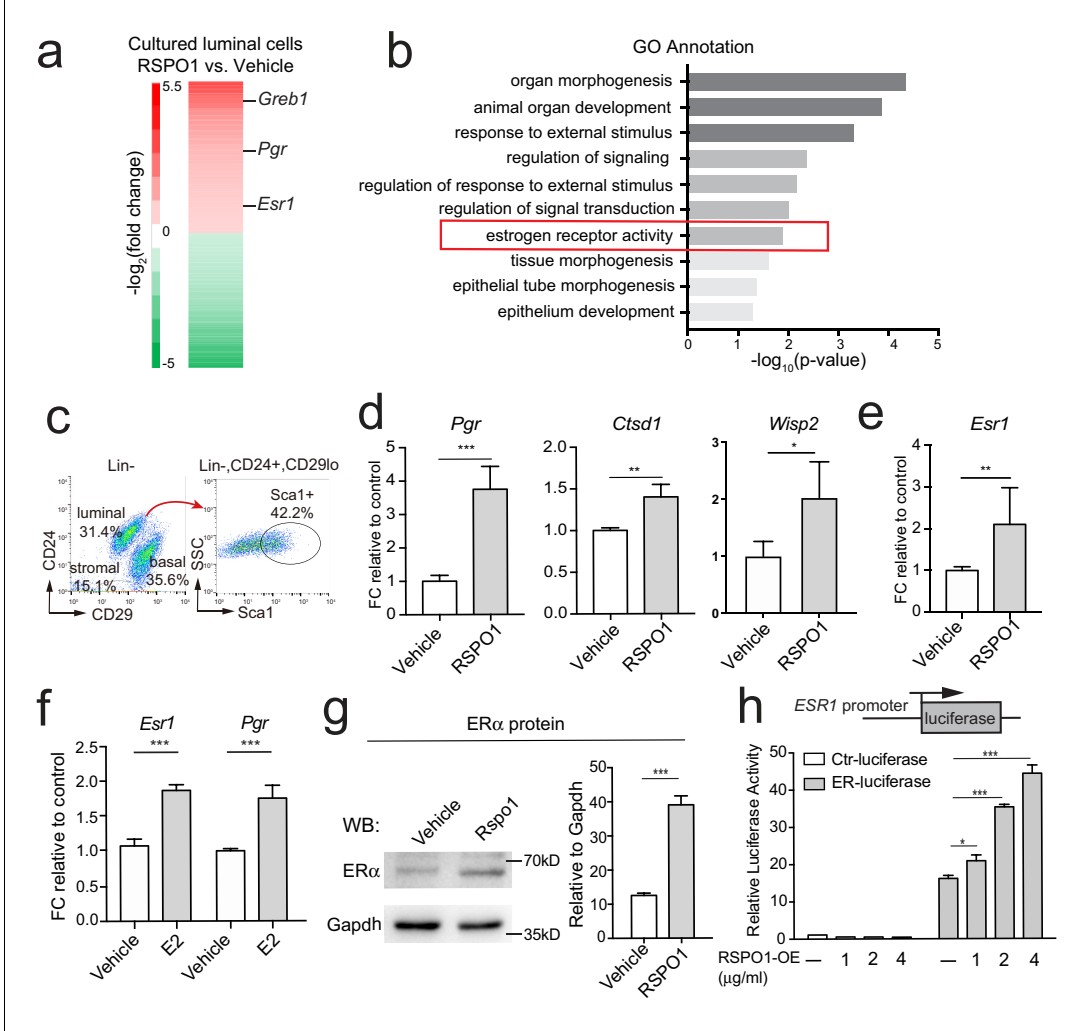

**Figure 1.** Rspo1 enhances Esr1 transcription and ERα signaling activities. (**a**) RNA-seq of 3D cultured luminal cells in the presence of RSPO1 (0.5 μg/ml) or vehicle. Increased expression of ERα target genes (*Pgr*, *Greb1*) and *Esr1* were enlisted in heatmap of differentially expressed genes (DEGs). (**b**) GO analysis was conducted on upregulated genes and estrogen receptor activity was enhanced in the presence of RSPO1. (**c**) Sca1[+] luminal cells were FACS-isolated. (**d, e**) qPCR analysis of cultured cells in day two indicating increased expression of *Esr1* (**e**) and its target genes (**d**) in the presence of RSPO1 (0.5 μg/ml). (**f**) E2 (1 μM) treatment was used as positive control indicating the upregulation of *Esr1* and its target *Pgr*. (**g**) Western analysis of cultured cells in day 2 showing increased ERα protein levels after RSPO1 treatment. (**h**) A luciferase reporter driven by the ESR1 promoter was constructed and transfected into HEK293T cells. RSPO1 treatment activated the *ESR1* promoter-luciferase reporter activities in a dose dependent manner. (**d–h**) Data are presented as mean ± s.e.m. of three independent experiments. Student's t test: \*\*\*p<0.001, \*\*p<0.01, \*p<0.05.

The online version of this article includes the following figure supplement(s) for figure 1:

**Figure supplement 1.** Rspo1 promotes ERα signaling activities.
**Figure supplement 2.** RSPO1 induces *Esr1* expression in a dose-dependent manner in Eph4 cell line.

hybridization validated the expression pattern of *Lgr4* in both basal and luminal layers (*Figure 2b*). We next investigated whether Lgr4 mediates Rspo1's action on *Esr1* expression. We generated Lgr4 shRNA and validated its knockdown efficacy in primary luminal cells by qPCR analysis (*Figure 2c*). Lgr4 knockdown suppressed the upregulation of *Esr1* induced by RSPO1 (*Figure 2d*). In an *ESR1*-luciferase reporter assay using T47D (a human breast cancer cell line), LGR4 knockdown also inhibited the luciferase activities induced by RSPO1 (*Figure 2e*). The effect was validated using two different shRNAs (*Figure 2e*, *Figure 2—figure supplement 1*). Results suggest that Rspo1 relies on Lgr4 to activate *Esr1* expression.

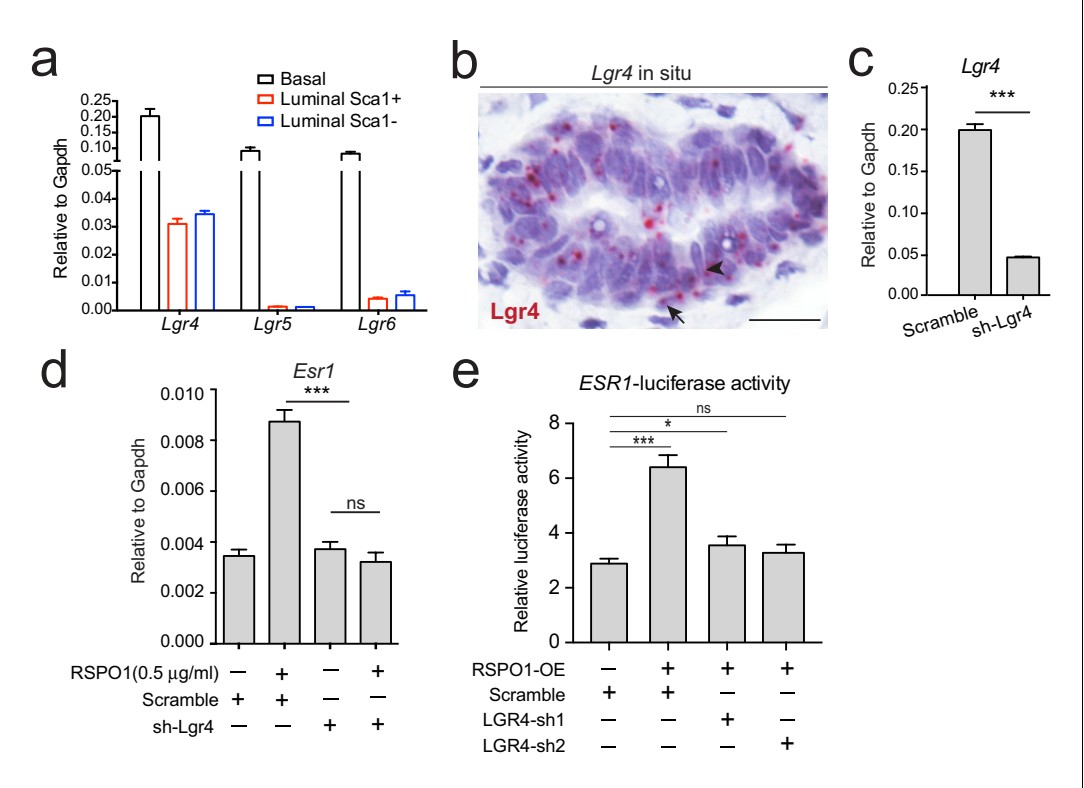

**Figure 2.** Rspo1 inducing *Esr1* expression is dependent on Lgr4. (**a**) qPCR analysis of Lgrs in FACS-isolated basal and luminal cells. *Lgr4*, *Lgr5* and *Lgr6* are all expressed in basal cells, while only *Lgr4* is distinctively expressed in luminal cells with even distribution in Sca1$^+$ (ER$^+$) and Sca1$^-$ (ER$^-$) luminal subpopulations. (**b**) Lgr4 in situ hybridization (in pink) confirming its expression in basal (arrow) and luminal cells (arrowhead). Nuclei were counterstained with hematoxylin (in purple). Scale bar, 20 μm. (**c**) qPCR analysis of *Lgr4* expression in cultured luminal cells indicating successful knockdown by shRNA. (**d**) qPCR analysis of *Esr1* expression in cultured luminal cells indicating that knockdown of *Lgr4* by shRNA counteracts the upregulation of *Esr1* by RSPO1. (**e**) *ESR1* promoter-luciferase reporter assays on T47D cells indicating that knockdown of LGR4 mRNA by shRNA counteracts the upregulation of *ESR1* by RSPO1, while scramble shRNA cannot. Data in (**c–f**) are pooled from three independent experiments and are presented as mean ± s.e.m. Student's t test: ***p<0.001, **p<0.01, *p<0.05; ns, not significant.

The online version of this article includes the following figure supplement(s) for figure 2:

**Figure supplement 1.** Validation of LGR4 shRNAs knockdown efficiency.

## ERα induction by Rspo1 is independent of Wnt/β-catenin signaling

As Rspo1 is known for amplifying Wnt/β-catenin signaling, we investigated whether Wnt ligands have a synergistic influence on *Esr1* expression. We first examined Wnt4, a major Wnt ligand in the mammary gland that can activate Wnt/β-catenin signaling (*Cai et al., 2014*; *Rajaram et al., 2015*). The activation of *Axin2* expression indicated that Wnt/β-catenin signaling was activated in primary luminal cell culture in the presence of Wnt4 (*Figure 3a*). Wnt4+RSPO1 combination further stimulated *Axin2* expression (*Figure 3a*). Intriguingly, addition of Wnt4 alone was ineffective in activating *Esr1* expression in these cells (*Figure 3b*), and Wnt4+RSPO1 combination was unable to further increase *Esr1* level compared to RSPO1 alone (*Figure 3b*). These results suggest that canonical Wnt signaling may not be involved in this regulatory axis. Furthermore, we used either Wnt3a or a GSK3 inhibitor CHIR99021 (CHIR) to stimulate Wnt/β-catenin signaling in primary luminal cell culture. Although Wnt-signaling activators markedly increased the expression levels of its target gene *Axin2* (*Figure 3c*), they could not stimulate *Esr1* expression (*Figure 3d*). It is noteworthy that the combination of RSPO1 with CHIR did not further induce *Axin2* level (*Figure 3c*), probably due to the Wnt/β-catenin signaling activity induced by CHIR or Wnt3a had reached plateau. In contrast to their

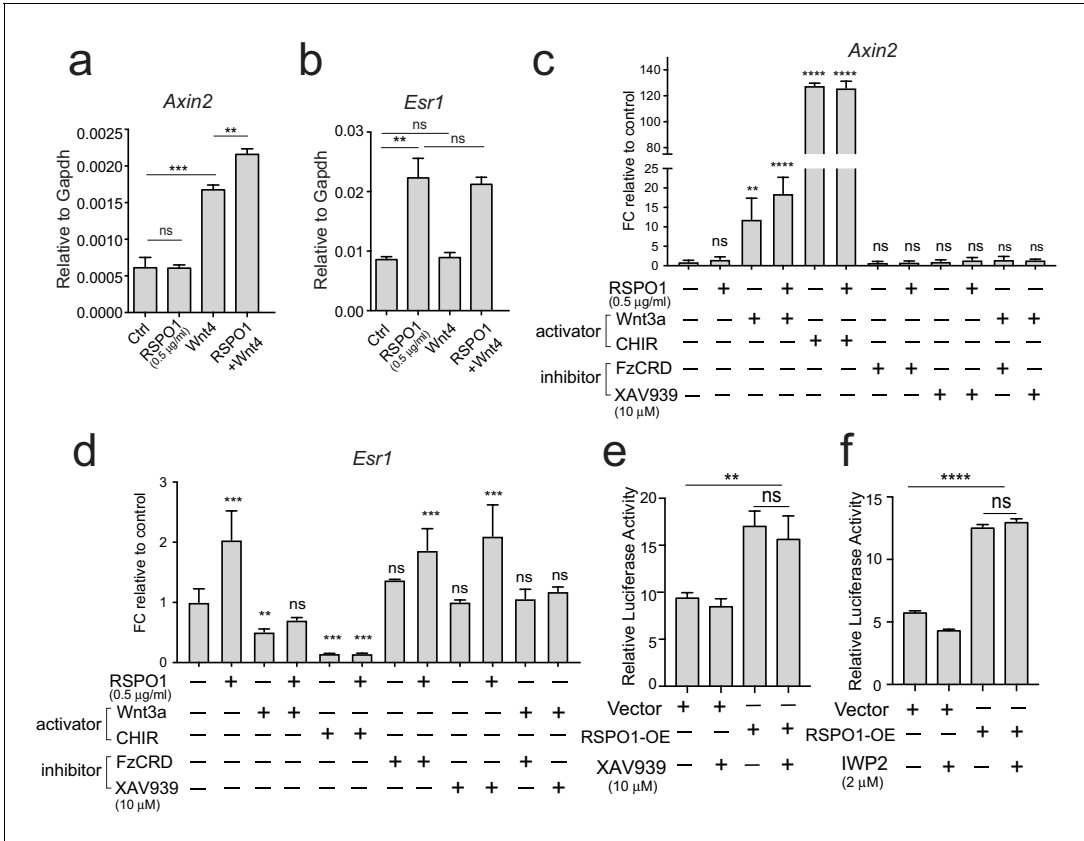

**Figure 3.** *Esr1* expression induced by Rspo1 is independent of Wnt/β-catenin signaling. (**a–b**) qPCR analysis of cultured luminal cells indicating that Wnt4 alone or in combination with RSPO1 can activate Wnt target *Axin2* expression (**a**). While RSPO1 alone promoted *Esr1* expression, Wnt4 was ineffective for *Esr1*. Combination of RSPO1 and Wnt4 displayed no difference compared with RSPO1 alone (**b**). (**c–d**) qPCR analysis of cultured luminal cells indicating that Wnt signaling activators (Wnt3a and GSK3β inhibitor CHIR) cannot activate *Esr1* expression, and that *Esr1* expression induced by RSPO1 cannot be suppressed by addition of Wnt signaling inhibitor (FzCRD or β-catenin inhibitor XAV939) (**d**). In contrast, Wnt-target gene *Axin2* expression was activated in the presence of Wnt signaling activators, and was suppressed by adding the signaling inhibitors (**c**). (**e–f**) HEK293T cells with transiently expressing *ESR1*-luciferase reporter were cultured in the presence of RSPO1, or in combination with Wnt inhibitors (XAV939 and IWP2). Wnt inhibitors cannot suppress *ESR1*-luciferase activities induced by RSPO1. Data in (**a–f**) are pooled from more than three independent experiments and presented as mean ± s.e.m. Student's t test. ****p<0.0001, ***p<0.001, **p<0.01, *p<0.05; ns, not significant.

stimulating effect to *Axin2*, Wnt3a and CHIR treatment suppressed *Esr1* expression (*Figure 3d*), an observation in line with a previous report, in which Wnt/β-catenin signaling represses the expression of luminal differentiation genes, mainly *Esr1* (*Lindley et al., 2015*). Inhibition of the Frizzled receptor using its soluble CRD domain (FzCRD) (*Hsieh et al., 1999*) or stimulating β-catenin degradation using XAV939 (*Huang et al., 2009*) effectively suppressed *Axin2* expression induced by Wnt3a (*Figure 3c*), still, they could not suppress *Esr1* upregulation by Rspo1 (*Figure 3d*). To further verify, we used HEK293T cells transiently expressing *ESR1*-luciferase reporter and cultured them in the presence of RSPO1 or RSPO1 in combination with XAV939 or IWP2. Consistently, inhibition of WNT signaling did not affect *ESR1* promoter activities induced by RSPO1 (*Figure 3e and f*). Together, these data suggest that Rspo1 induces ERα expression independent of Wnt/β-catenin signaling.

## Loss of luminal Rspo1 results in decreased ERα expression in vivo

To investigate the role of Rspo1 in vivo, we generated a conditional *Rspo1* knockout allele in which the second *Rspo1* exon is subjected to removal upon *Cre* recombination, resulting in frame-shift of the remaining exons (*Figure 4a*, also see *Figure 4—figure supplement 1a–b*). Of note, *Rspo1* is predominantly expressed in ER- luminal cells as described previously (*Cai et al., 2014*), while *Esr1* is expressed in ER+ luminal cells. Thus, this Rspo1-*Esr1* regulation is likely achieved through a paracrine

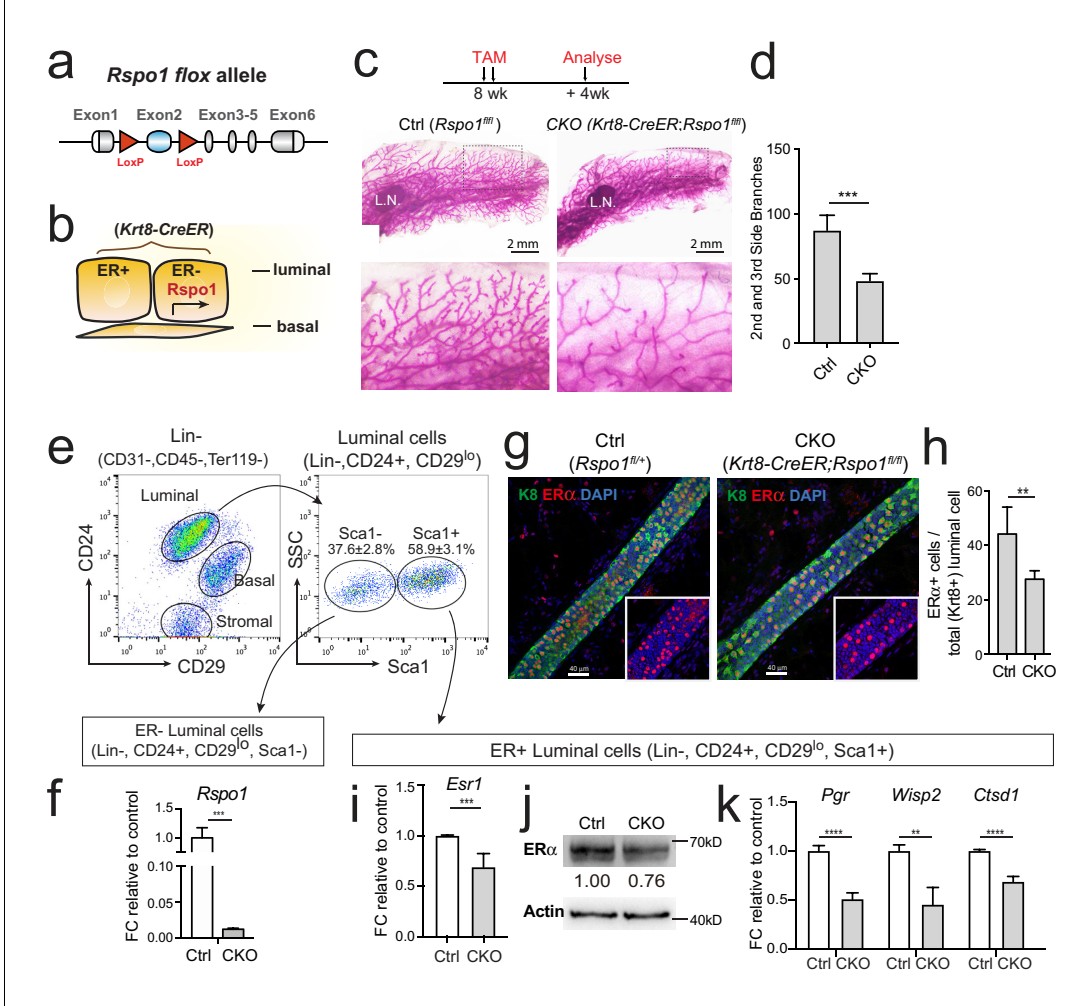

**Figure 4.** Loss of Rspo1 in mammary luminal cells results in reduced side branching and decreased ERα expression. (a) Schematic illustration of *Rspo1^flox^* knock-in allele generation (see also *Figure 4—figure supplement 1*). (b) *Krt8-CreER;Rspo1^fl/fl^* inducible model specifically knockdown Rspo1 in luminal cells. (c–d) 8-week-old adult virgin mice were Tamoxifen injected twice, 1 day apart (2 mg/25 g body weight per injection). Mammary glands were obtained 4 weeks later. Whole-mount imaging (c) of mammary epithelium and quantification (d) showing decreased side branches in Rspo1-cKO mice. n = 3. Scale bar, 2 mm. More than six views were used for quantification. (e–f) FACS gating strategy for mammary basal and luminal cell isolation. Luminal ER⁺ and ER⁻ subpopulations were separated based on Sca1 (e). qPCR analyses of luminal cells showing efficient Rspo1 knockdown in *Krt8-CreER;Rspo1^fl/fl^* (f). (g–h) Immunostaining indicated decreased ERα⁺ cell number after Rspo1 knockdown (g). Scale bar, 40 μm. Quantification of ERα⁺ cells were performed in (h). (i) qPCR analyses of ER⁺ luminal cells indicated downregulation of *Esr1* transcription after Rspo1 knockdown in ER⁻ cells. (j) Sca1⁺ (ER⁺) luminal cells were FACS isolated and Western blot was performed to indicate decreased ERα expression after Rspo1 knockdown. (k) qPCR analyses of ER⁺ luminal cells indicated downregulation of ERα target genes after Rspo1 knockdown in ER⁻ cells. Data are presented as mean ± s.e.m. of more than three independent experiments. Student's t test: ****p<0.0001, ***p<0.001, **p<0.01, *p<0.05. L.N. Lymph node.

The online version of this article includes the following figure supplement(s) for figure 4:

**Figure supplement 1.** Generation of *Rspo1^flox^* mouse model.

**Figure supplement 2.** Loss of *Lgr4* reduces *Esr1* expression and ERα signaling activities.

manner in vivo. A luminal cells-specific BAC transgenic CreER line, *Keratin8-CreER* (*Krt8-CreER*) (*Zhang et al., 2012a*), was used to generate luminal cells-specific *Rspo1* knock-out mice (*Krt8-CreER;Rspo1^fl/fl^*) (Rspo1-cKO) (*Figure 4b*). Tamoxifen was administered into 8-week-old nulliparous female mice, and mammary glands were examined 4 weeks later. Whole-mount carmine staining showed significantly reduced side branches in Rspo1-cKO mice when compared with the control (*Rspo1^fl/fl^*) (*Figure 4c and d*). These results are consistent with previous observation in a *Rspo1^-/-^* mammary transplantation model (*Chadi et al., 2009*). The knockout efficacy of Rspo1-cKO was validated. ER⁻ luminal cells (Lin⁻, CD24⁺, CD29^lo^, Sca1⁻), where Rspo1 is expressed, were isolated

(*Figure 4e*). *Rspo1* level in cKO group was significantly reduced shown by qPCR analysis (*Figure 4f*). By whole-mount immunofluorescence staining, we observed the decreased ERα expression in *Rspo1-cKO* mammary gland (*Figure 4g*). Quantification indicated decreased percentage of ERα$^+$ cells (*Figure 4h*), likely reflecting the overall reduction of ERα level in luminal compartment. Although we could not exclude the possible switching of ER$^+$ to ER$^-$ cell fate due to other indirectly reasons, we tested a more direct possibility—whether it is the reduction of ERα expression in ER+ compartment that results in loss of ER$^+$ cells. To this end, we isolated ER+ luminal cells (Lin$^-$, CD24$^+$, CD29$^{lo}$, Sca1$^+$), and analyzed ERα levels as well as ERα signaling activities. We found that ERα levels were reduced in this compartment as shown by qPCR (*Figure 4i*) and Western analysis (*Figure 4j*). Consistently, ERα signaling target genes, including *Pgr*, *Wisp2* and *Ctsd1* were declined in *Rspo1-cKO* group (*Figure 4k*). Therefore, together these results suggest that loss of Rspo1 results in reduced ERα expression and its signaling activities in luminal cells.

The *Esr1* expression was also examined in *Lgr4$^{lacZ}$* mouse model, a hypomorphic allele of Lgr4, (*Mazerbourg et al., 2004*). Mammary glands of Lgr4 homozygous mutant (*Lgr4$^{lacZ/lacZ}$*) were isolated for whole mount imaging. At 9 weeks, *Lgr4$^{lacZ/lacZ}$* mammary glands displayed significantly less side branches (*Figure 4—figure supplement 2a–b*). Immunostaining revealed decreased ERα expression in *Lgr4$^{lacZ/lacZ}$* mammary gland (*Figure 4—figure supplement 2c–d*). When ER$^+$ luminal cells (Lin$^-$, CD24$^+$, CD29$^{lo}$, Sca1$^+$) were isolated, we found that *Esr1* was significantly reduced in *Lgr4* mutant, so were the ERα downstream targets *Pgr*, *Ctsd1* and *Wisp2* (*Figure 4—figure supplement 2e*). *Lgr4* expression was markedly decreased in *Lgr4$^{lacZ/lacZ}$* mammary gland as a validation of the hypomorphic nature of the allele (*Figure 4—figure supplement 2e*). These results support that Lgr4 plays a role in mediating Rspo1-induced ERα expression.

## Genetic evidence supports that *Esr1* regulation is independent of luminal Wnt4

To investigate whether Wnt/β-catenin signaling affects *Esr1* in vivo, we also generated a Wnt4 conditional knockout mouse. In this model, the second *Wnt4* exon is flanked by flox, and is removed upon *Cre* recombination, which leads to frame shift of the remaining exons (*Figure 5a*, *Figure 5—figure supplement 1a–b*). We generated inducible, luminal cells-specific *Wnt4* knock-out mice (*Krt8-CreER; Wnt4$^{fl/fl}$*) (*Figure 5b*). Tamoxifen was administered into 8-week-old nulliparous female mice, and mammary glands were examined 4 weeks later. Loss of Wnt4 resulted in reduced side branching (*Figure 5c and d*), consistent with a previous report using *MMTV-Cre;Wnt4$^{fl/fl}$* model (*Rajaram et al., 2015*).

To address whether loss of Wnt4 affects *Esr1*, we isolated ER$^+$ luminal populations from both *Wnt4*-cKO (*Krt8-CreER;Wnt4$^{fl/fl}$*) and control (*Krt8-CreER;Wnt4$^{fl/+}$*) mammary gland. qPCR and Western analyses both indicated that loss of Wnt4 increases ERα level in ER$^+$ luminal cells (*Figure 5f–g*), as well as ERα signaling activities showed by increased target gene expression (*Figure 5h*). These were in contrast to the reduced *Esr1* level and ERα signaling activity observed in Rspo1-cKO mice (*Figure 4i–k*). These were consistent with the in vitro results that Wnt3a and CHIR treatment suppressed *Esr1* expression (*Figure 3d*), and consistent with the previous report, in which Wnt-controlled transcriptional regulator LBH repress luminal genes, mainly *Esr1* (*Lindley et al., 2015*). The successful deletion of Wnt4 in cKO group was validated by significantly reduced *Wnt4* level in ER$^+$ luminal cells (*Figure 5e*), as well as reduced expression of Wnt/β-catenin signaling targets *Axin2* and *Lgr5* in basal cells (*Figure 5i*). Together, in vivo genetic evidence supports that *Esr1* regulation is independent of luminal Wnt4.

## Rspo1 relies on cAMP-PKA pathway to induce *Esr1* expression

To further investigate the downstream mechanisms through which Rspo1/Lgr4 regulate *Esr1*, we conducted an inhibitor-based screen. HEK293T cells with transiently expressing *ESR1*-luciferase reporter were cultured in the presence of RSPO1, and screened for molecules that could suppress luciferase activity using a GPCR inhibitor library (*Figure 6a*, *Figure 6—figure supplement 1*, *Figure 6—source data 1*). Amongst over 250 inhibitors, the cAMP inhibitor Bupivacaine HCl (Bup), effectively suppressed *ESR1*-luciferase activities induced by RSPO1 (*Figure 6a*). Considering that the major downstream effector of cAMP in mammalian cells is Protein Kinase A (PKA), we examined the effect of inhibition of PKA. Consistently, H89, an inhibitor of PKA effectively repressed *ESR1*-

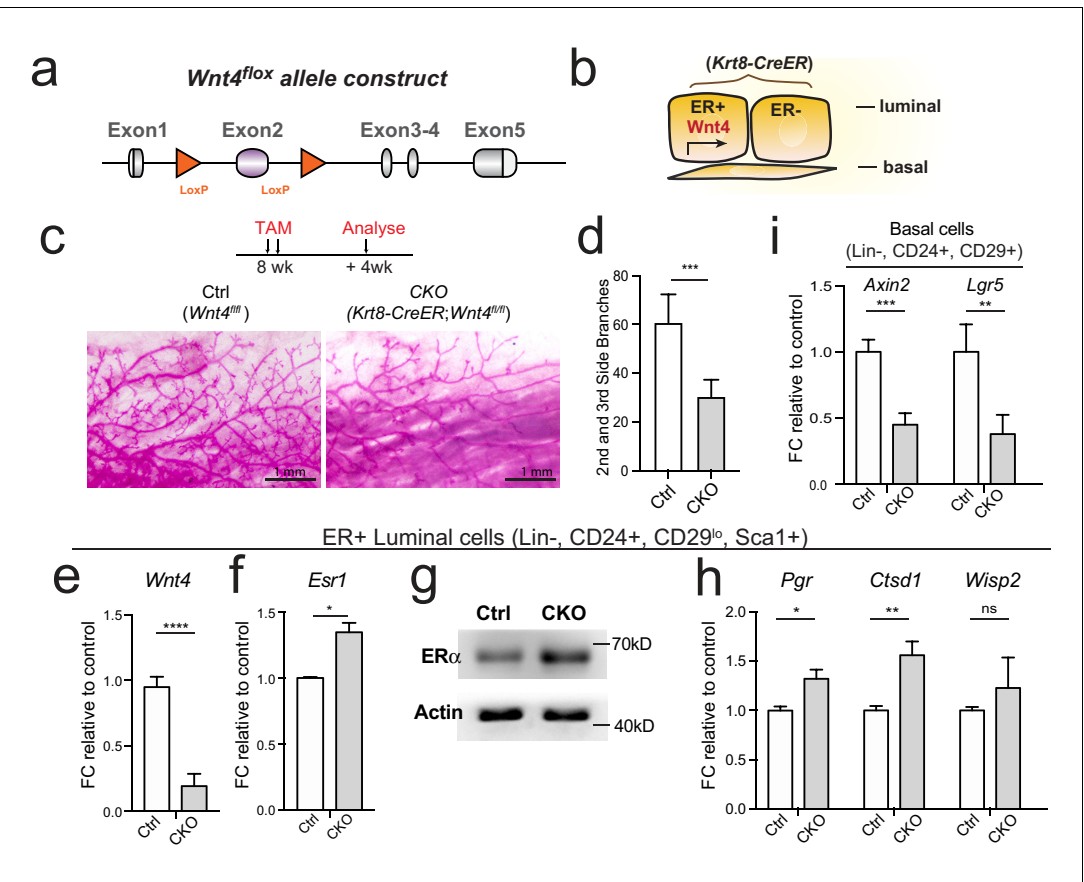

**Figure 5.** Loss of Wnt4 increases *Esr1* expression in luminal cells. (a) Schematic illustration of *Wnt4flox* knock-in allele generation (see also ***Figure 5—figure supplement 1***). (b) *Krt8-CreER;Wnt4flox* inducible model specifically knockdown Wnt4 in luminal cells. (c–d) 8-week-old adult virgin mice were Tamoxifen administered for 2 courses at 1 day apart, 2 mg/25 g body weight per injection and harvested 4 weeks later. Whole-mount imaging of the mammary epithelium showing decreased side branches in *Wnt4*-cKO mice (c). n = 3. Scale bar, 1 mm. More than six views were used for quantification. (e) qPCR of isolated ER⁺ luminal cells validated efficient Wnt4 knockdown in cKO mice. (f) qPCR analysis of ER⁺ luminal cells indicated Wnt4 loss increased *Esr1* expression levels. (g) Western blot analysis indicated increased ERα protein level in *Wnt4*-cKO. (h) qPCR analysis of ER⁺ luminal cells indicated increased ERα signaling pathway activity after Wnt4 knockdown. (i) qPCR analysis of basal cells showed Wnt signaling pathway was decreased after Wnt4 knockdown. Data are presented as mean ± s.e.m. Student's t test: ****p<0.0001, ***p<0.001, **p<0.01, *p<0.05; ns, not significant.

The online version of this article includes the following figure supplement(s) for figure 5:

**Figure supplement 1.** Generation of *Wnt4flox* mouse model.

luciferase activities induced by RSPO1 (***Figure 6a***). The inhibitory effects of Bup and H89 were further examined in primary luminal cell culture. Both inhibitors suppressed *Esr1* expression stimulated by Rspo1 as shown by qPCR (***Figure 6b***), but were ineffective on *Axin2* expression (***Figure 6c***). Considering the cAMP-PKA pathway can also be activated by estrogen and ERα (***Castoria et al., 2008***), we further examined whether *Esr1* induction by RSPO1 involves ERα. We found that the ERα inhibitor ICI (ICI182, 780) does not affect *ESR1* promoter activities that are induced by RSPO1 (***Figure 6d***), suggesting that *Esr1*-induction by Rspo1 does not involve ERα.

The transcription factor CREB (cAMP response element binding protein) is the best-characterized nuclear protein that mediates stimulation of transcription by cAMP. CREB binds to the conserved consensus cAMP response element (CRE, sequence TGACATCA) (***Rosenberg et al., 2002***). A CRE was found at the proximal promoter of *ESR1* (−991 to −984 bp). Therefore, we examined whether this CRE is responsible for induction of *ESR1* by RSPO1. While RSPO1 induced the wild type promoter-luciferase in a dose-dependent manner, it could not activate the reporter with CRE mutations

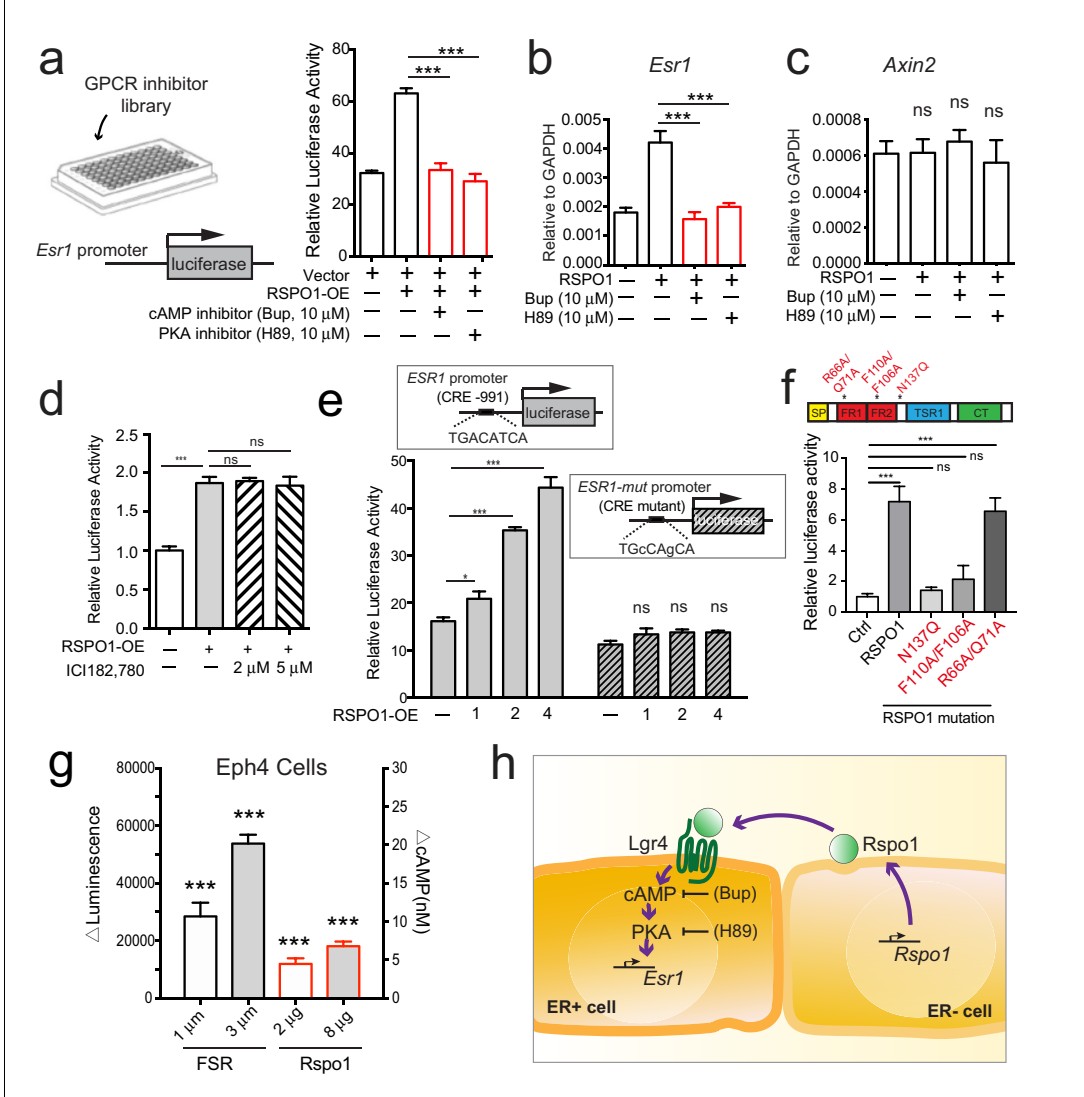

**Figure 6.** Rspo1 inducing *Esr1* expression relies on cAMP-PKA pathway. (a) HEK293T cells transfected with *ESR1*-luciferase reporter were cultured in the presence of RSPO1, and in combination with pharmaceutical compounds from a GPCR inhibitor library (Selleck). Bupivacaine HCl (Bup), a cAMP inhibitor, and H89, a PKA inhibitor, suppressed *Esr1*-luciferase activities induced by RSPO1 (see *Figure 6—figure supplement 1a–b*). (b–c) qPCR analysis of cultured luminal cells indicating both Bup and H89 counteracted the upregulation of *Esr1* expression induced by RSPO1 (b), while *Axin2* expression is not affected (c). (d) HEK293T cells with transiently expressing *ESR1*-luciferase reporter were cultured in the presence of RSPO1 alone or with ERα inhibitor ICI182, 780. Luciferase activities were measured. ICI did not affect the *ESR1* upregulation induced by RSPO1. (e) CRE site on *ESR1* promoter-luciferase reporter was mutated, and RSPO1 could not activate the reporter with CRE mutation. (f) RSPO1-FL, RSPO1-R66A/Q71A mutant could, but RSPO1-N137Q and RSPO1-F110A/F106A mutants could not induce *Esr1* promoter luciferase activities. (g) ΔLuminescence was read out after Eph4 cells were treated with forskolin (FSR) or RSPO1 for 30 min, ΔLuminescence was calculated as Luminescencetreated—Luminescenceuntreated. Rspo1 treatment induced cAMP production in Eph4 cells in a dose dependent manner. (h) Illustration of Rspo1 regulated Esr1 expression mediated by cAMP-PKA pathway. Data in (a–d) are pooled from three independent experiments and presented as mean ± s.e.m. Student's t test: ***p<0.001, **p<0.01, *p<0.05; ns, not significant.

The online version of this article includes the following source data and figure supplement(s) for figure 6:

**Source data 1.** ESR1-luciferase activities induced by RSPO1 in combination with a GPCR inhibitor library.

**Figure supplement 1.** *ESR1*-luciferase activities induced by RSPO1 in combination with a GPCR inhibitor library HEK293 cells with transiently expressing ESR1-luciferase reporter were cultured in the presence of RSPO1, and screened for the molecules that could suppress the luciferase activities using a GPCR inhibitor library (Selleck, L2200).

(TGcCAgCA) (*Figure 6e*), rendering its specificity. In addition, we co-expressed *ESR1*-luciferase with RSPO1-FL (full length), RSPO1-N137Q (a mutated form with compromising secretion) (*Chang et al., 2016*), RSPO1 F110A/F106A (unable to bind LGR4) (*Wang et al., 2013*), and RSPO1 R66A/Q71A (binds to LGRs but is unable to amplify Wnt signaling) (*Xie et al., 2013*). RSPO1-FL and RSPO1 R66A/Q71A were able to activate the luciferase activities, but RSPO1-N137Q and RSPO1 F110A/ F106A could not (*Figure 6f*), suggesting the secretion of Rspo1 and its association to LGR4 are critical for *Esr1* transcription. This is in line with the paracrine mechanism we propose. Next, we attempted to directly measure the change of cAMP level upon Rspo1 treatment in mammary Eph4 cells, using Forskolin (FSR) as a positive control. We observed a dose dependent increase of cAMP level in relation to increasing RSPO1 stimulation (*Figure 6g*). Together, these results suggest that Rspo1 signals through cAMP-PKA-CREB axis to promote *Esr1* transcription (illustrated in *Figure 6h*).

## Discussion

In this study, we uncover a novel function and signaling mechanism of Rspo1 in promoting ERα expression. This action of Rspo1 is dependent on Lgr4 and G-protein coupled cAMP/PKA pathway, but independent of Wnt/β-catenin signaling. In vivo, the biological significance of this regulatory axis is first revealed in the mammary gland homeostasis. Luminal cells-specific deletion of Rspo1 results decreased ERα expression and reduced side branching.

### A novel Wnt-independent role of Rspo1

Rspo1 has been known as a stem cell growth factor in many adult tissues with prominent biological and therapeutic significance. The action of Rspo1 on stem cells is through strongly potentiating Wnt signaling (*Carmon et al., 2011*; *de Lau et al., 2011*; *Glinka et al., 2011*; *Gong et al., 2012*; *Hao et al., 2012*; *Koo et al., 2012*). Here we unveil a new role of Rspo1 in promoting *Esr1* transcription in hormone receptor-positive luminal cells. This Wnt/β-catenin-independent action of Rspo1 relies on the Lgr4 receptor and intracellular cAMP/PKA signaling. Knockdown of Lgr4 counteracts Rspo1's augmenting effect on ERα transcription in vitro and in vivo, while modulation of Wnt input or β-catenin activity does not affect *Esr1* level induced by Rspo1. Same holds true in vivo when using Wnt4-cKO mouse model. Deletion of *Wnt4* in luminal cells increased *Esr1* level. To our best knowledge, this is the first report of Wnt/β-catenin independent function of Rspo1 in physiological condition. It adds to previous reported Wnt/β-catenin independent role of Rspo1 in antagonizing colon cancer metastasis, in which LGR5 directly binds to TGFβ receptors for the activation of TGFβ signaling (*Zhou et al., 2017*).

### Rspo1 activates G-protein coupled cAMP signaling in regulating *Esr1*

cAMP is a well-known intracellular mediator of protein hormones including FSH (follicle-stimulating hormone), LH (luteinizing hormone), and TSH (thyroid stimulating hormone), which bind to LGR1, LGR2 and LGR3 respectively (*de Lau et al., 2014*). These known hormone receptors belong to the class-A LGRs. Class-B LGRs, including LRG4-6, are reported to promote phosphorylation of Lrp5/6 and stabilization of β-catenin without the G-protein-coupled cAMP production (*Carmon et al., 2011*; *de Lau et al., 2011*). There have been a few reports that suggest differently, in that Lgr4 activates cAMP/PKA signaling in bone (*Luo et al., 2009*), and in the male reproductive system (*Li et al., 2010*). Independently, our data demonstrate that Rspo1/Lgr4 relies on the cAMP/PKA axis to maintain proper *Esr1* expression during mammary development. This action is highly likely cell type specific. In vivo, either conditional KO of Rspo1 or Lgr4 hypomorphic mutant leads to reduced *Esr1* expression. The latter is in line with previous reports in the male reproductive system, in which deficiency of Lgr4 results in reduced *Esr1* in the efferent ducts and epididymis (*Hoshii et al., 2007*; *Li et al., 2010*). The current study, for the first time, demonstrates that Rspo1 can activate cAMP/ PKA signaling.

### A new hormonal regulation feed forward mechanism

Our previous studies find that hormones indirectly activate Rspo1 expression in ER⁻ luminal cells (*Cai et al., 2014*), and identify Areg (in ER⁺ cells) as the intermediate paracrine factor for the hormonal regulation of Rspo1 expression (in ER⁻ cells) (*Cai et al., 2020*). Moreover, the elevated levels of Areg and Rspo1 are also detected in estrus, a stage with high estrogen signaling activity

(*Cai et al., 2020*). In this study, we found that Rspo1 in turn enhances ERα expression in ER$^+$ cells. This may represent a feed forward mechanism engaging estrogen-ERα-Rspo1-ERα, highlighting the impact of local growth factors for the amplification of hormonal signaling output. This additional layer of ERα regulation by Rspo1 could be hijacked during tumor initiation or progression. Elucidating the molecular mechanisms on how estrogen engages with ERα in the mammary gland is the key for advancing current knowledge over breast cancer progression and resistance to hormone therapy.

In conclusion, our study demonstrated a novel Wnt-independent role of Rspo1, revealed a novel Rspo1-Lgr4-cAMP-ERα regulatory axis. As ERα is crucial for the development and diseases of various tissues, this new Rspo1 signaling axis may have broader implication in estrogen-associated diseases.

# Materials and methods

## Key resources table

| Reagent type (species) or resource | Designation | Source or reference | Identifiers | Additional information |
|---|---|---|---|---|
| Genetic reagent (*M. musculus*) | *Krt8-CreERT2* | PMID:22350718 | RRID:MGI:5314229 | Dr. Li Xin (Department of Molecular and Cellular Biology, Baylor College of Medicine, United States) |
| Genetic reagent (*M. musculus*) | *Rspo1^{flox/+}* | This paper | | Generated in our laboratory Detail refer to *Figure 4* and *Figure 4—figure supplement 1* |
| Genetic reagent (*M. musculus*) | *Wnt4^{flox/+}* | This paper | | Generated in our laboratory Detail refer to *Figure 5* and *Figure 5—figure supplement 1* |
| Genetic reagent (*M. musculus*) | *Lgr4^{LacZ/+}* | PMID:15192078 | RRID:MGI:3052121 | Drs. Minyao Liu and Dali Li |
| Cell line (*M. musculus*) | Eph4 | PMID:25260709 | | Mouse mammary epithelial cell line |
| Cell line (*H. sapiens*) | HEK293T (293T) | http://www.cellbank.org.cn | Cat. #: SCSP-502 | |
| Cell line (*H. sapiens*) | T47D | Dr. Gaoxiang Ge's laboratory | | Human breast cancer cell line Dr. Gaoxiang Ge (Institute of Biochemistry and Cell Biology) |
| Antibody | Rabbit anti Gapdh GAPDH Polyclonal Antibody | Proteintech | Cat. #: 10494–1-AP RRID:AB_2263076 | WB (1:3000) |
| Antibody | Mouse anti β-Actin Monoclonal Antibody | Sigma | Cat. #: A2228 RRID:AB_476697 | WB (1:2000) |
| Antibody | Rabbit anti ERα Polyclonal Antibody | Millipore | Cat. #: 06–935 RRID:AB_310305 | WB (1:1000), IHC (1:200) |
| Antibody | Rat anti Krt8 Monoclonal Antibody | DSHB | Cat. #: TROMA-1 | IHC (1:500) |
| Antibody | Rat Anti-Mouse CD31 Monoclonal Antibody, Biotin Conjugated | BD PharMingen | Cat. #: 553371 RRID:AB_394817 | Flow cytometry (1:200) |
| Antibody | Rat Anti-Mouse CD45 Monoclonal Antibody, Biotin Conjugated | BD PharMingen | Cat. #: 553078 RRID:AB_394608 | Flow cytometry (1:200) |

*Continued on next page*

*Continued*

| Reagent type (species) or resource | Designation | Source or reference | Identifiers | Additional information |
|---|---|---|---|---|
| Antibody | Rat Anti-Mouse TER-119 Monoclonal Antibody, Biotin Conjugated | BD PharMingen | Cat. #: 553672 | Flow cytometry (1:200) |
| Antibody | Rat Anti-Mouse CD31 Monoclonal Antibody, FITC Conjugated | BD PharMingen | Cat. #: 553372 RRID:AB_394818 | Flow cytometry (1:200) |
| Antibody | Rat Anti-Mouse CD45 Monoclonal Antibody, FITC Conjugated | BD PharMingen | Cat. #: 553080 RRID:AB_394610 | Flow cytometry (1:200) |
| Antibody | Rat Anti-Mouse TER119 Monoclonal Antibody, FITC Conjugated | BD PharMingen | Cat. #: 557915 | Flow cytometry (1:200) |
| Antibody | PE/Cy7 Rat Anti-Mouse CD24 Monoclonal Antibody | Biolegend | Cat. #: 101–822 RRID:AB_756048 | Flow cytometry (1:200) |
| Antibody | APC Armenian Hamster Anti-Mouse/Rat CD29 Monoclonal Antibody | Biolegend | Cat. #: 102216 RRID:AB_492833 | Flow cytometry (1:200) |
| Antibody | Rat Anti-Ly-6A/E (Sca-1) Monoclonal Antibody, PE | eBioscience | Cat. #: 12-5981-82 RRID:AB_466086 | Flow cytometry (1:200) |
| Antibody | Streptavidin eFluor 450 Conjugate | eBioscience | Cat. #: 48-4317-82 RRID:AB_10359737 | Flow cytometry (1:200) |
| Peptide, recombinant protein | Epithelial growth factor (EGF) | Corning | Cat. #: 354001 | 50 ng/mL |
| Peptide, recombinant protein | FzCRD | PMID:25260709 | | Inhibition of the Frizzled receptor using its soluble CRD domain. Purified in our laboratory (1:200) |
| Peptide, recombinant protein | Wnt3A | PMID:20569694 | | Purified in our laboratory 200 ng/ml |
| Recombinant DNA reagent | plko.1 backbone | Addgene | RRID:Addgene_30323 | |
| Recombinant DNA reagent | pGL4.17 basic vector | Promega, Madison, WI, U.S.A. | Cat. #: E6721 | |
| Recombinant DNA reagent | pRL-TK Renilla | Promega, Madison, WI, U.S.A. | Cat. #: E2241 | |
| Recombinant DNA reagent | pcDNA3.1-RSPO1 overexpression (RSPO1-OE) plasmid | This paper | | Constructed in our laboratory Detail refer to Materials and methods |
| Chemical compound, drug | tamoxifen (TAM) | Sigma-Aldrich | Cat. #: T5648 | 2 mg/20 g |
| Chemical compound, drug | HEPES | Sigma | Cat. #: H4034-500G | 25 mM |
| Chemical compound, drug | Collagenase III | Worthington | Cat. #: LS004183 | 300 U/ml |
| Chemical compound, drug | red blood cell lysing buffer | Sigma | Cat. #: R7757 | |
| Chemical compound, drug | DNase I | Sigma | Cat. #: D4263 | 0.1 mg/mL |
| Chemical compound, drug | Carmine | Sigma | Cat. #: C1022 | 2 mg/ml |
| Chemical compound, drug | Histoclear | National Diagnostics | Cat. #: HS-200 | |

*Continued on next page*

*Continued*

| Reagent type (species) or resource | Designation | Source or reference | Identifiers | Additional information |
|---|---|---|---|---|
| Chemical compound, drug | Matrigel | BD Bioscience | Cat. #: 354230 | |
| Chemical compound, drug | Insulin-Transferrin-Selenium (ITS) | Gibco | Cat. #: 41400045 | Cell culture (1:100) |
| Chemical compound, drug | E2 | Sigma | Cat. #: E8875 | 1 μM |
| Chemical compound, drug | IWP2 | Selleck | Cat. #: s7085 | 2.5 μM |
| Chemical compound, drug | CHIR | Selleck | Cat. #: S1263 | 3 μM |
| Chemical compound, drug | XAV-939 | Selleck | Cat. #: S1180 | 10 μM |
| Chemical compound, drug | Protein A Agarose | Santa Cruz | Cat. #: sc-2003 | |
| Chemical compound, drug | DAPI | Life Technologies | Cat. #: P36931 | |
| Chemical compound, drug | GPCR compound library | Selleckchem L2200 | | Chemical Biology Core Facility, Institute of Biochemistry and Cell Biology, SIBS, CAS |
| Commercial assay or kit | In situ hybridization RNAscope kit | Advanced Cell Diagnostics | | Following the manufacturer's instructions |
| Commercial assay or kit | Dual-Luciferase Reporter Assay System | Promega | Cat. #: E1910 | Following the manufacturer's instructions |
| Commercial assay or kit | cAMP-Glo assay kit | Promega | Cat. #: V1501 | Following the manufacturer's instructions |
| Commercial assay or kit | SuperScript III kit | Invitrogen | Cat. #: RR036A | Following the manufacturer's instructions |
| Software, algorithm | GraphPad Prism | GraphPad Prism (https://graphpad.com) | | |
| Software, algorithm | ImageJ | ImageJ (http://imagej.nih.gov/ij/) | | |

## Experimental animals

*Rspo1*$^{flox/+}$ and *Wnt4*$^{flox/+}$ mice were constructed as illustrated in the text. In all conditional knockout experiments, mice were maintained on a C57BL/6 genetic background and at least three animals were analyzed for each genotype. *Lgr4*$^{LacZ/+}$(**Mazerbourg et al., 2004**) and *Krt8-CreERT2* (**Zhang et al., 2012a**) strains were used in this study. Nude, CD1 and BALB/c strains were purchased from B and K universal (Shanghai). Animals were housed under conditions of 12 h day/night cycle.

For Cre recombination induction experiments induced in adult mice, animals received intraperitoneal injection of 2 mg tamoxifen (TAM; Sigma-Aldrich; T5648) diluted in sunflower oil. The Animal Care and Use Committee of Shanghai Institute of Biochemistry and Cell Biology, Chinese Academy of Sciences approved experimental procedures.

## Antibodies

Rabbit anti Gapdh (1:3000; Proteintech; 10494–1-AP), Mouse anti β-Actin (1:2000; Sigma; A2228) and Rabbit anti ERα (1:1000; Millipore; 06–935) were used in Western blot analyses.

## Primary cell preparation

Mammary glands from 8- to 12-wk-old virgin female mice were isolated. Minced tissues were placed in digestion buffer (RPMI 1640 [Gibco; C11875500BT] with 25 mM HEPES [Sigma; H4034-500G], 5% FBS [Hyclone], 1% PSQ [Gibco; 15140122], 300 U mL$^{-1}$ Collagenase III [Worthington; LS004183]) and digested for 2 hr at 37°C. After lysis of the red blood cells in red blood cell lysing buffer (Sigma;

R7757), a single cell suspension was obtained by sequential incubation with 0.05% Trypsin-EDTA (Gibco; 25300–062) for 5 min at 37°C and 0.1 mg/mL DNase I (Sigma; D4263) for 5 min with gentle pipetting followed by filtration through 70 μm cell strainers (Falcon; 352350).

## RNA extraction and RNA sequencing

Total RNA from day two cultured luminal cells (Lin⁻, CD24+, CD29$^{lo}$) were extracted with RNAiso Plus (Takara) following manufacturer's protocol. Total mRNA concentration was determined with NanoDrop ND-1000 and RNA-seq libraries were prepared according to manufacturer's instruction (Illumina) followed by applying to sequencing on Illumina nova-seq, which was performed by ANOR-OAD (http://en.annoroad.com, Beijing). Differential gene expression analysis was carried out and genes with significant alteration were extracted and further analysed using DAVID Bioinformatics Resources. RNA-seq data can be viewed online at http://www.biosino.org/node/index, under accession number OEP000754.

## Mammary gland whole mount carmine staining

The 4th pair of mammary glands were dissected and fixed for 2 hr in 4% paraformaldehyde, and then washed the tissue three times in PBS for 15 min each time. Finally, the tissues were stained in carmine alum solution (2 mg/ml carmine [Sigma; C1022], 5 mg/ml $KAl(SO_4)_2$ in $H_2O$) overnight at room temperature. After the staining, the tissues were washed in de-staining solution (50% ethanol, 2% HCl) for 2 hr, and then serial dehydrated in 75%, 85%, 95%, 100%, 100% ethanol and finally stored in Histoclear (National Diagnostics; HS-200). Whole mount analyses were performed under a dissection microscope (Leica).

## Mammary gland whole mount immunostaining

Whole-mount staining was performed as previously described (*Rios et al., 2014*), with minor modification. In brief, mammary glands were dissected into small pieces, then processed in digestion buffer (RPMI 1640 with 25 mM HEPES, 5% fetal bovine serum, 1% penicillin–streptomycin–glutamine (PSQ), 300 U/ml collagenase III (Worthington)) for 30 min at 37°C, then fixed in 4% paraformaldehyde for 30 min at 4°C. Tissues were incubated with primary antibodies (Krt8; 1:500; DSHB, ER; 1:200; Millipore) at 4°C overnight, followed by washes, incubated with secondary antibodies and DAPI (Life Technologies) at 4°C overnight. Then the tissues were incubated in 80% glycerol overnight, before dissection for 3D imaging. Confocal images were captured using Leica SP8 laser confocal scanning microscope. Representative images were shown in the figures.

## Cell labeling and flow cytometry

The following antibodies in 1:200 dilutions were used: biotinylated and FITC conjugated CD31, CD45, and TER119 (BD PharMingen; 553371; 553078; 553672; 553372; 553080; 557915); CD24-PE/cy7, CD29-APC (Biolegend; 101–822; 102216) Sca1-PE and Streptavidin-V450 (eBioscience; 12-5981-82; 48-4317-82). Antibody incubation was performed on ice for 25 min in PBS with 5% FBS. All sorting experiments were performed using a FCAS Jazz (Becton Dickinson). The purity of sorted population was routinely checked and ensured to be >95%.

## In vitro culture assay

FACS-sorted cells were resuspended in chilled 100% growth factor-reduced Matrigel (BD Bioscience; 354230), and the mixture was allowed to polymerize before covering with culture medium (DMEM/F12 [Gibco; 11039–021]; ITS [1:100; Gibco; 41400045]; 50 ng mL⁻1 EGF [Corning; 354001]), plus either 1 μM E2 (Sigma; E8875), 200 ng Wnt3A, 1:100 FzCRD, 2.5 μM IWP2 (Selleck; s7085), 3 μM CHIR (Selleck; S1263), 10 μM XAV-939 (Selleck; S1180), Rspo1 purified protein or Wnt4 conditioned media. Culture medium was changed every 24 hr. Cell samples were collected after 2–4 days in culture for RT-qPCR and western blot.

## Maintenance of cell lines

293T and Eph4 cell lines were cultured in DMEM high glucose (4.5 g/L) (Gibco, C11995500BT) with 1% Penicillin/Streptomycin (Gibco, 15140) and 10% Fetal bovine serum (FBS) (Hyclone). Both cell lines were cultured in tissue culture dish, kept at 37°C with 5% $CO_2$, trypsinized, and split three times

a week 1:4. T47D cell line was kindly provided by Dr. Gaoxiang Ge, Institute of Biochemistry and Cell Biology and was cultured in 1640 Medium (Gibco, C11875500BT) +10 mg/ml Insulin with 1% Penicillin/Streptomycin and 10% FBS. All cell lines were routinely negatively tested for mycoplasma.

## Conditioned media preparation

Wnt4 conditional medium was prepared by culturing Wnt4-expressing Eph4 cells for 48 hr, followed by supernatant collect. Wnt4 conditional medium was stored at 4°C for short-term storage (up to 1 week). For long-term usage, conditional medium was aliquoted after collection and stored at −80°C.

## RSPO1 protein purification

RSPO1-FC construct was cloned into expression vector with a C-terminal Fc tag. RSPO1-FC was transiently expressed in HEK293T cells and medium changed into CD293 medium (Gibco, 11913–019). One day after transfection, medium was collected by centrifugation and incubated with Protein A Agarose Beads (Santa Cruz, sc-2003). The bound recombinant protein was eluted using 500 μl 0.1M Glycine (pH = 3.0) and was collected in 1.5 ml tubes containing 30 ul 1 M Tris-HCl (pH = 9.5) buffer for neutralization. In total 5 tubes of elution were collected. The RSPO1 protein was subsequently purified and concentrated by Centrifugal Filter Volumes (Millipore, UFC803096).

## Lentiviral vector and infection

Lgr4-shRNA was synthesized and subcloned into plko backbone with EGFP. Lentivirus was produced by transient transfection in 293 T cells. Mammary cells were isolated from 8- to 12-wk-old virgin female glands as described above, followed by sorting into luminal cells. The sorted cells were collected and cultured in a low adherent plate in EGF, ITS-supplemented DMEM/F12 with virus. At 12 hr after infection, cells were collected and resuspended in Matrigel for consequent in vitro culturing. Sequences of Lgr4-shRNA are CGTAATCAAATCTCCCTGATA and CCTCCAGAACAATCAGTTGAA.

## Luciferase assay

Oligonucleotide primers (nucleotides −1133 to −1107 and −1 to −24 based on previously published sequence information for the upstream region of the *ESR1* were used to generate *ESR1* promoter fragments from normal placental DNA by polymerase chain reaction (PCR) (*Castles et al., 1997*). A 1133 bp (promoter A) of *ESR1* promoter expression vector (ERP) was created by cloning this PCR-generated product into the *XhoI-HindIII* sites of the promoterless luciferase reporter plasmid pGL4.17 basic respectively (Promega, Madison, WI, U.S.A.). Transfections of individual wells were performed using luciferase reporter plasmid (ERP or pGL4.17 basic vector alone), and pRL-TK Renilla luciferase control constructs as a correction for transfection efficiency, and also transfected with pcDNA3.1-RSPO1 overexpression (RSPO1-OE) plasmid (from 0.5 μg/ml to 4 μg/ml)) Cells were then harvested, the dual luciferase assays were performed using a commercial kit (Promega; E1910), Results are shown as fold activity over control activity of the promoterless pGL4.17 basic vector in each set of experiments. All transfections and assays were performed in duplicate with n ≥ 3 individual experiments. GPCR compound library (Selleckchem L2200) was used to for screening of inhibitors that suppress *ESR1* upregulation by RSPO1. In each experiment, *ESR1*-lucieferase reporter cells were treated with RSPO1 for 36–48 hr.

## In situ hybridization

In situ hybridization was performed using the RNAscope kit (Advanced Cell Diagnostics) following the manufacturer's instructions. *Lgr4* probes were ordered from Advanced Cell Diagnostics. For in situ staining, at least three independent experiments were conducted. Representative images are shown in the figures.

## AMP-Glo assay to detect intracellular cAMP levels

The intracellular cAMP concentration was measured using the cAMP-Glo assay kit (Promega, V1501) according to the manufacturer's instruction. The cAMP standard curve was generated using purified cAMP, from which the relative intracellular level of cAMP was inferred. For each drug treatment, three biological repeats were used, and each experiment was repeated 2–3 times.

## RT-qPCR

RNA was isolated with Trizol (Invitrogen; 9109). The cDNA library was prepared with the SuperScript III kit (Invitrogen; RR036A). RT–PCR was performed on a StepOne Plus (Applied Biosystems). RNA level was normalized to GAPDH. The primers used were as following:

Axin2-F, AGCCTAAAGGTCTTATGTGGCTA;
Axin2-R, ACCTACGTGATAAGGATTGACT;
Wnt4-F, GCAATTGGCTGTACCTGG;
Wnt4-R, GCACTGAGTCCATCACCT;
Rspo1-F, GCAACCCCGACATGAACAAAT;
Rspo1-R, GGTGCTGTTAGCGGCTGTAG;
Esr1-F, TCCAGCAGTAACGAGAAAGGA
Esr1-R, AGCCAGAGGCATAGTCATTGC
Pgr-F, GGGGTGGAGGTCGTACAAG
Pgr-R, GCGAGTAGAATGACAGCTCCTT
Lgr4-F, AGAACTCAAAGTCCTAACCCTC
Lgr4-R, ATGCCGCAACTGAACGAG
Lgr5-F, CCTACTCGAAGACTTACCCAGT
Lgr5-R, GCATTGGGGTGAATGATAGCA
Lgr6-F, CTGTAGCCCTGGTGATGA
Lgr6-R, GGGTTGAAGAGCAGGTAG
Ctsd1-F, GCTTCCGGTCTTTGACAACCT
Ctsd1-R, CACCAAGCATTAGTTCTCCTCC
Wisp2-F, TGTGTGACCAGGCAGTGATG
Wisp2-R, GTGCTCCAGTTTGGACAGGG.

## Statistical analysis

One-way ANOVA or Student's t-test was performed, and the P-value was calculated in Prism on data represented by bar charts, which consisted of results from three independent experiments unless otherwise specified. For all experiments with error bars, the standard deviation (SD) was calculated to indicate the variation within each experiment. No statistical method was used to predetermine sample size. The experiments were not randomized. The investigators were not blinded to allocation during experiments and outcome assessment.

## Acknowledgements

We are grateful to Drs. Minyao Liu and Dali Li for kindly sharing of Lgr4$^{-/-}$ mice, to Dr. Esther Verheyen for critical reading of the manuscript and Dr. Chi-Chung Hui for helpful suggestion. This work was supported by the Ministry of Science and Technology of China (2019YFA0802002 to YAZ), the National Natural Science Foundation of China (31625020, 31530045, 31830056, 31861163006 to YAZ; 81873532 to QCY; 31671546 to CC), Chinese Academy of Sciences (XDB19020200, XDA16020200 to YAZ) QCY gratefully acknowledges the support of SA-SIBS Scholarship Program.

## Additional information

### Funding

| Funder | Grant reference number | Author |
|---|---|---|
| National Natural Science Foundation of China | 31625020 | Yi Arial Zeng |
| Chinese Academy of Sciences | XDB19020200 | Yi Arial Zeng |
| National Natural Science Foundation of China | 31530045 | Yi Arial Zeng |
| National Natural Science Foundation of China | 31830056 | Yi Arial Zeng |
| National Natural Science Foundation of China | 31861163006 | Yi Arial Zeng |

| | | |
|---|---|---|
| National Natural Science Foundation of China | 81873532 | Qing Cissy Yu |
| National Natural Science Foundation of China | 31671546 | Cheguo Cai |
| National key research and development program of China | 2019YFA0802002 | Yi Arial Zeng |
| Chinese Academy of Sciences | XDA16020200 | Yi Arial Zeng |

The funders conceived the study, wrote the manuscript and made the decision to submit the work for publication.

## Author contributions

Ajun Geng, Data curation, Investigation, Visualization; Ting Wu, Validation, Investigation, Visualization; Cheguo Cai, Investigation; Wenqian Song, Validation, Methodology; Jiqiu Wang, Resources; Qing Cissy Yu, Supervision, Project administration, Writing - review and editing; Yi Arial Zeng, Conceptualization, Writing - original draft, Writing - review and editing

## Author ORCIDs

Ajun Geng ⓘ https://orcid.org/0000-0001-7456-6566
Qing Cissy Yu ⓘ https://orcid.org/0000-0001-7516-7137
Yi Arial Zeng ⓘ https://orcid.org/0000-0003-1898-8099

## Ethics

Animal experimentation: The Animal Care and Use Committee of Shanghai Institute of Biochemistry and Cell Biology, Chinese Academy of Sciences approved experimental procedures (SIBCB-S335-1601-002-c4).

## Decision letter and Author response

Decision letter https://doi.org/10.7554/eLife.56434.sa1
Author response https://doi.org/10.7554/eLife.56434.sa2

# Additional files

## Supplementary files

• Transparent reporting form

## Data availability

RNA-seq data can be viewed online at http://www.biosino.org/node/index, under accession number OEP000754.

The following dataset was generated:

| Author(s) | Year | Dataset title | Dataset URL | Database and Identifier |
|---|---|---|---|---|
| Geng A | 2020 | A Novel Function of R-spondin1 | http://www.biosino.org/node/index | OEP000754, OEP000754 |

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
