## [Decision Letter]

**Acceptance summary:**

We are very excited to highlight your studies revealing a novel non-Wnt role for Rspo1 via cAMP-PKA activation. We believe that your studies will expand our understanding of how components of the Wnt pathway act through other non-related pathways.

**Decision letter after peer review:**

Thank you for submitting your article "A novel function of R-spondin1 in regulating estrogen receptor expression independent of Wnt/β-catenin signaling" for consideration by *eLife*. Your article has been reviewed by three peer reviewers, and the evaluation has been overseen by Edward Morrisey as the Senior and Reviewing Editor. The following individuals involved in review of your submission have agreed to reveal their identity: Christof Niehrs (Reviewer #1); Stijn De Langhe (Reviewer #3).

The reviewers have discussed the reviews with one another and the Reviewing Editor has drafted this decision to help you prepare a revised submission.

As the editors have judged that your manuscript is of interest, but as described below that additional experiments are required before it is published, we would like to draw your attention to changes in our revision policy that we have made in response to COVID-19 (https://elifesciences.org/articles/57162). First, because many researchers have temporarily lost access to the labs, we will give authors as much time as they need to submit revised manuscripts. We are also offering, if you choose, to post the manuscript to bioRxiv (if it is not already there) along with this decision letter and a formal designation that the manuscript is 'in revision at *eLife*'. Please let us know if you would like to pursue this option. (If your work is more suitable for medRxiv, you will need to post the preprint yourself, as the mechanisms for us to do so are still in development.)

In particular, please pay attention to the comments by reviewer 1 and comments 1-3 by reviewer 2. While some of these will require additional data, others may be addressed be a careful editing of the conclusions made in the manuscript.

Reviewer #1:

The manuscript by Geng et al., by the Zeng lab uncovers a new role of R-Spondin 1 (Rspo1) for estrogen receptor (*Esr1*) expression in mouse mammary gland. Rspo1 has been exclusively considered as a potent agonist in WNT signaling, where it plays a crucial role in stem cell maintenance and development. Geng and co-authors discovered an unexpected WNT-independent role of Rspo1 in cAMP-PKA signaling to regulate transcription of *Esr1*. The data support a paracrine mechanism involving an Rspo1-*Lgr4*-cAMP-PKA-*Esr1* axis between mammary gland ER+ and ER- cells.

Overall, this study is nicely designed and executed with a battery of biochemical analyses further supported by genetic evidence in mouse mutants. This study has broad significance for the mammary development, estrogen signaling, Rspo-Lgr, and Wnt signaling fields.

My only main point concerns the major finding that Rspo1 regulates the cAMP-PKA-*Esr1* axis via *Lgr4*. To better support this conclusion, the authors should provide some additional data:

1) Figure 4H: For Rspo1 cKO, the authors show Pgr, *Wisp2*, *Ctsd1* and *Cyp1b1* expressions but for the *Lgr4* mutant they only show Pgr and Greb1 (Supplementary Figure 4C). The authors should also check *Wisp2*, *Ctsd1* and *Cyp1b1* expressions in *Lgr4* mutant, to test if these markers show the same trend of Rspo1 cKO, to better demonstrate that Rspo1 and *Lgr4* cooperates for regulating *Esr1* transcription.

2) Figure 4: The authors should perform a western blot or immunofluorescence of ERα in *Lgr4* lof cells, as they did with Rspo1 cKO (4i,g).

Reviewer #2:

Dr. Zeng's group previously reported that estrogen and progesterone signaling could activate RSPO1 and that RSPO1 and *Wnt4* synergize to promote mammary stemness. In this manuscript, the authors studied the effect of RSPO1 on modulating *Esr1* transcription and ERα signaling. They observed that RSPO1 can induce *Esr1* expression in cultured mammary cells, and went on to show that RSPO1 affects *Esr1* expression in mammary cells through LGR4-cAMP-PKA-CREB but independent of Wnt signaling. Demonstrating a Wnt signaling-independent effect on *Esr1* transaction is significant as RSPO1 is primarily known to bind LGR4/5/6 to potentiate Wnt signaling. However, LGR4 has been reported previously to activate *Esr1* in non-mammary cells and via cAMP-PKA-CREB cells as cited in the Discussion. Furthermore, this work is largely preliminary based on cultures of cells isolated from mice and primarily one single mouse cell line Eph4. The direct in vivo evidence is weak. Therefore, while it is important to study how RSPO1 regulates *Esr1* in mammary cells, this manuscript did not make a significant further advance beyond what is already published. Main concerns are listed below:

1) The in vivo evidence that RSPO1 regulates *Esr1* expression in mammary cells is weak. The authors generated a Rspo1 conditional KO mouse line. In comparing the *Esr1* expression between WT and KO mice, the authors showed a tangential cut of a duct rather than a typical duct with lumen (Figure 4G), and they failed to quantify and compare these immunofluorescence-stained ducts. Such quantification would have provided much needed in vivo evidence. Instead, the authors chose to isolate a subset of mammary cells for qPCR for ERα transcriptional targets and ERα Western blotting. However, RSPO1 may have changed the cell fate, and the isolated cells from the KO mice could be a skewed subset, unsuitable for comparison with the WT cell preparations. Furthermore, the ERα Western lacked statistical consideration, and *Esr1* qPCR was not performed.

2) To support their claim that RSPO1 regulates *Esr1* independent of Wnt signaling, they presented in vitro data (Figure 3) that suppressing Wnt signaling could not dampen RSPO1 effects on *Esr1* expression while activating Wnt signaling suppressed *Esr1* expression. Next, they generated a tamoxifen induced knockout line of *Wnt4* (Figure 5). It is bewildering that the authors did not carefully examine the ER expression in the mammary glands from these mice. Rather they only performed qPCR analysis of a subset of mammary cells isolated from conditional *Wnt4* KO mice, which suffers the same flaw as above. The qPCR indeed showed decreased *Esr1* expression, but the Figure 5 legend states that "Loss of *Wnt4* does not affect *Esr1* expression in luminal cells," leading this reviewer to wonder what exactly the authors attempted to prove. Furthermore, any observation of knockout of a single member of the large Wnt gene family on *Esr1* expression cannot be extrapolated to support the claim that RSPO1 activation of *Esr1* expression does not require Wnt signaling. In addition, it is a huge stretch to cite Lindley et al., 2015 to claim that Wnt/catenin has been reported to suppress ER when the cited paper only showed LBH can regulate ER. LBH is one of the hundreds of target genes that may be regulated by Wnt signaling in various contexts. Not every function of these huge family of target genes can be extrapolated as Wnt signaling effects.

3) In addition to RSPO1-N137Q in Figure 6F, the authors should test other RSPO1 mutants in inducing *Esr1* expression, including RSPO1 F110A/F106A (which could not bind to LGR4) and RSPO1 R66A/Q71A (which could bind to LGR4 but could not amplify Wnt signaling). These experiments are important as positive data would provide direct evidence that LGR4 but not activation of Wnt/β-catenin signaling is required for RSPO1 in inducing *Esr1* expression.

4) The authors presented evidence that RSPO1 can modulate cAMP-PKA-CREB to transactivate *Esr1*; however, the authors cannot make the jump that RSPO1 does this via LGR4 and independent of Wnt signaling. These authors did not test in their own cell lines whether LGR4 is required for this RSPO1 regulation of cAMP-PKA-CREB and whether the G protein activities of LGR4 are involved in *Esr1* expression.

5) While it is reasonable to isolate ERα positive mammary cells for testing the effect of RSPO1, the data would have been much more convincing if they also included ERα-negative cells as a control and showed no induction of ERα. The authors also did not explain why RSPO1 needs to act by a paracrine manner to activate LGR4 to activate *Esr1* expression. What prevents RSPO1 from activating LGR4 on the RSPO1-producing cells?

6) Only one shRNA against LGR4 is used. It is uncertain that the observed effects on *Esr1* are not due to off-target targets. This is especially grave in Figure 2E which shows that this LGR4 shRNA can suppress *Esr1* in MCF7 cells, which has been reported to have very low levels of LGR4.

7) It is unclear whether RSPO1 affects Wnt signaling in their experiment models. In Figure 3A, RSPO1 treatment of mammary cells in vitro did not affect *Axin2* mRNA levels. However, in Figure 3C, RSPO1 alone significantly enhanced *Axin2* mRNA levels. There is no explanation for this discrepancy. The authors did not show whether their conditional RSPO1 knockout mice impacted Wnt signaling.

Reviewer #3:

This is an elegantly performed study demonstrating a novel function of R-spondin1 in regulating estrogen receptor expression independent of Wnt/β-catenin signaling.

I have no concerns.

---

## [Author Response]

[…] My only main point concerns the major finding that Rspo1 regulates the cAMP-PKA-Esr1 axis via Lgr4. To better support this conclusion, the authors should provide some additional data:1) Figure 4H: For Rspo1 cKO, the authors show Pgr, Wisp2, Ctsd1 and Cyp1b1 expressions but for the Lgr4 mutant they only show Pgr and Greb1 (Supplementary Figure 4C). The authors should also check Wisp2, Ctsd1 and Cyp1b1 expressions in Lgr4 mutant, to test if these markers show the same trend of Rspo1 cKO, to better demonstrate that Rspo1 and Lgr4 cooperates for regulating Esr1 transcription.

Following the reviewer’s suggestion, we performed qPCR analysis for *Wisp2, Ctsd1* and *Cyp1b1* and have included the data into revised Figure 4—figure supplement 2E. Consistent with the decrease seen in Rspo1-cKO, expression of *Wisp2* and *Ctsd1* also reduced in *Lgr4* mutant. We did not included *Cyp1b1*, as its expression level was very low, therefore its change may not be a proper indication of ERα signaling activities. For similar reason, we also removed *Cyp1b1* data from revised Figure 4K.

2) Figure 4: The authors should perform a western blot or immunofluorescence of ERα in Lgr4 lof cells, as they did with Rspo1 cKO (Figure 4G, I).

Following the suggestion, we performed immunofluorescent staining of ERα in control (*Lgr4^+/-^*) and *Lgr4^-/-^* mutant mammary sections (revised Figure 4—figure supplement 2C, D). As expected, the proportion of ERα+ luminal cell was decreased in *Lgr4^-/-^*, in line with the decrease seen in Rspo1-cKO.

Reviewer #2:Dr. Zeng's group previously reported that estrogen and progesterone signaling could activate RSPO1 and that RSPO1 and Wnt4 synergize to promote mammary stemness. In this manuscript, the authors studied the effect of RSPO1 on modulating Esr1 transcription and ERα signaling. They observed that RSPO1 can induce Esr1 expression in cultured mammary cells, and went on to show that RSPO1 affects Esr1 expression in mammary cells through LGR4-cAMP-PKA-CREB but independent of Wnt signaling. Demonstrating a Wnt signaling-independent effect on Esr1 transaction is significant as RSPO1 is primarily known to bind LGR4/5/6 to potentiate Wnt signaling. However, LGR4 has been reported previously to activate Esr1 in non-mammary cells and via cAMP-PKA-CREB cells as cited in the Discussion. Furthermore, this work is largely preliminary based on cultures of cells isolated from mice and primarily one single mouse cell line Eph4. The direct in vivo evidence is weak. Therefore, while it is important to study how RSPO1 regulates Esr1 in mammary cells, this manuscript did not make a significant further advance beyond what is already published. Main concerns are listed below:1) The in vivo evidence that RSPO1 regulates Esr1 expression in mammary cells is weak. The authors generated a Rspo1 conditional KO mouse line. In comparing the Esr1 expression between WT and KO mice, the authors showed a tangential cut of a duct rather than a typical duct with lumen (Figure 4G), and they failed to quantify and compare these immunofluorescence-stained ducts. Such quantification would have provided much needed in vivo evidence. Instead, the authors chose to isolate a subset of mammary cells for qPCR for ERα transcriptional targets and ERα Western blotting. However, RSPO1 may have changed the cell fate, and the isolated cells from the KO mice could be a skewed subset, unsuitable for comparison with the WT cell preparations. Furthermore, the ERα Western lacked statistical consideration, and Esr1 qPCR was not performed.

We respectfully disagree with the reviewer. The wholemount immunostaining shown in Figure 4G (the reviewer referred as "a tangible cut") provides a broader view compared to the traditional cross section. In our opinion, whole mount immunostaining is a technical advance that better demonstrates the distribution/density of ERα+ cells along the mammary duct, which cross section fails to achieve. Following the suggestion, we included quantification data showing that decreased percentage of ERα+ cells (revised Figure 4H).

I hope the reviewer agrees that, based on the above observation, performing qPCR or WB using total luminal cells would merely confirm the decreased percentage of ERα+ cells, instead of providing further information.

In our opinion, the above quantification likely reflects the overall reduction of ERα level in luminal compartment. Although we could not exclude the possible switching of ER+ to ER- cell fate due to other indirect reasons, we tested a more direct possibility—whether it is the reduction of ERα expression in ER+ compartment that results in loss of ER+ cells. To this end, we isolated ER+ luminal cells (Lin^-^, CD24^+^, CD29^lo^, Sca1^+^), and analyzed ERα levels as well as ERα signaling activities. We were able to show that on top of decreased percentage of ERα+ cells, the remaining ERα+ cells also displayed reduced ERα level and compromised ERα signaling activities (Figure 4I-K). The loss of ERα+ cells (Figure 4G-H) and the reduction in ERα level in the remaining ERα+ cells (Figure 4J) likely reflect different extents of Rspo1 regulation over ERα+ expression.

Following the suggestion, we have included the qPCR analysis of *Esr1* (revised Figure 4I).

2) To support their claim that RSPO1 regulates Esr1 independent of Wnt signaling, they presented in vitro data (Figure 3) that suppressing Wnt signaling could not dampen RSPO1 effects on Esr1 expression while activating Wnt signaling suppressed Esr1 expression. Next, they generated a tamoxifen induced knockout line of Wnt4 (Figure 5). It is bewildering that the authors did not carefully examine the ER expression in the mammary glands from these mice. Rather they only performed qPCR analysis of a subset of mammary cells isolated from conditional Wnt4 KO mice, which suffers the same flaw as above. The qPCR indeed showed decreased Esr1 expression, but the Figure 5 legend states that "Loss of Wnt4 does not affect Esr1 expression in luminal cells," leading this reviewer to wonder what exactly the authors attempted to prove. Furthermore, any observation of knockout of a single member of the large Wnt gene family on Esr1 expression cannot be extrapolated to support the claim that RSPO1 activation of Esr1 expression does not require Wnt signaling. In addition, it is a huge stretch to cite Lindley et al., 2015 to claim that Wnt/catenin has been reported to suppress ER when the cited paper only showed LBH can regulate ER. LBH is one of the hundreds of target genes that may be regulated by Wnt signaling in various contexts. Not every function of these huge family of target genes can be extrapolated as Wnt signaling effects.

We respectfully disagree with the reviewers. As explained above, using isolated ER+ luminal cells to examine ER signaling is a more stringent experimental design in our opinion. But we understand the reviewer’s concern, thus we also performed qPCR using whole luminal cells from control and *Wnt4*-cKO mice, which confirmed that loss of *Wnt4* increases *Esr1* expression and ER signaling activities (see Author response image 1).

Our apologies that the title in Figure 5 caused confusion. We have modified it to “Loss of *Wnt4* increases *Esr1* expression in luminal cells”.We respectfully disagree with the reviewers regarding the issue of overstatement when describing the effect of *Wnt4*. In Figure 3, when we concluded that “*Esr1* expression induced by Rspo1 is independent of Wnt/β-catenin signaling”, aside from *Wnt4*, we used activators Wnt3a and CHIR, and inhibitors FzCRD and XAV939. Those results collectively supported our conclusion. Subsequently, in Figure 5, we used *Wnt4*-cKO as a genetic model for in vivo validation, considering that *Wnt4* is the major Wnt member in adult mammary epithelium, and that *Wnt4*/Rspo1 synergy has been reported in the mammary gland. In our opinion, our subtitle, “Genetic evidence supports that *Esr1* regulation is independent of luminal *Wnt4*” was precise, and did not extend to refer other Wnt/β-catenin signaling.

Regarding how to cite the Lindley et al., 2015 paper, it has been rephrased to “…consistent with the previous report, in which Wnt-controlled transcriptional regulator LBH repress luminal genes, mainly *Esr1* (Lindley et al., 2015).” Of note, “Wnt-controlled transcriptional regulator LBH” is the exact word used in the title of Lindley et al., 2015.

3) In addition to RSPO1-N137Q in Figure 6F, the authors should test other RSPO1 mutants in inducing Esr1 expression, including RSPO1 F110A/F106A (which could not bind to LGR4) and RSPO1 R66A/Q71A (which could bind to LGR4 but could not amplify Wnt signaling). These experiments are important as positive data would provide direct evidence that LGR4 but not activation of Wnt/β-catenin signaling is required for RSPO1 in inducing Esr1 expression.

We thank the reviewer for the suggestion, and performed the experiments accordingly. As shown in revised Figure 6F, both RSPO1-N137Q mutant (prevents RSPO1 secretion) and RSPO1-F110A/F106A mutant (unable to bind to LGR4) abolished RSPO1's ability to activate *Esr1* luciferase reporter, while treatment with RSPO1-R66A/Q71A mutant (could not amplify Wnt signaling) showed similar *Esr1* luciferase activation as treatment with wildtype RSPO1. These results further supported that Wnt/β-catenin signaling is not required for *Esr1*-induction by Rspo1.

4) The authors presented evidence that RSPO1 can modulate cAMP-PKA-CREB to transactivate Esr1; however, the authors cannot make the jump that RSPO1 does this via LGR4 and independent of Wnt signaling. These authors did not test in their own cell lines whether LGR4 is required for this RSPO1 regulation of cAMP-PKA-CREB and whether the G protein activities of LGR4 are involved in Esr1 expression.

We apologize for the confusion caused. The requirement for *Lgr4* in Rspo1 induced *Esr1* expression has been shown in primary luminal cells (Figure 2D) and in T47D cells (Figure 2E). The connection of Rspo1 to cAMP-PKA-CREB-*Esr1* is revealed in Figure 6 as the reviewer pointed out. The connection of *Lgr4* to cAMP-PKA is known^1-3^. Together, in our opinion, it is logical to propose such a signaling model.

5) While it is reasonable to isolate ERα positive mammary cells for testing the effect of RSPO1, the data would have been much more convincing if they also included ERα-negative cells as a control and showed no induction of ERα. The authors also did not explain why RSPO1 needs to act by a paracrine manner to activate LGR4 to activate Esr1 expression. What prevents RSPO1 from activating LGR4 on the RSPO1-producing cells?

We thank the reviewer for the suggestion. We isolated ER- luminal cells and cultured them in the presence of RSPO1 for 2 days. Treatment of RSPO1 to ER- luminal cells didn’t result in significant change in *Esr1* expression. We have included this result in revised Figure 1—figure supplement 1C.

The reviewer raised an interesting point. At this stage, we don’t know what prevents Rspo1 from acting on ER- cells. It could be the intrinsic mechanism of ER- cells, which is a subject of future study.

6) Only one shRNA against LGR4 is used. It is uncertain that the observed effects on Esr1 are not due to off-target targets. This is especially grave in Figure 2E which shows that this LGR4 shRNA can suppress Esr1 in MCF7 cells, which has been reported to have very low levels of LGR4.

Following the suggestions, we generated one additional shRNA and replaced MCF7 cells with T47D cells that have higher LGR4 expression. We confirmed that both shRNAs (sh1 and sh2) are efficient in reducing endogenous *LGR4* expression in T47D cells (revised Figure 2—figure supplement 1), and produce consistent results in eliminating RSPO1 induced *Esr1*-luciferase reporter activities (Revised Figure 2E).

7) It is unclear whether RSPO1 affects Wnt signaling in their experiment models. In Figure 3A, RSPO1 treatment of mammary cells in vitro did not affect Axin2 mRNA levels. However, in Figure 3C, RSPO1 alone significantly enhanced Axin2 mRNA levels. There is no explanation for this discrepancy. The authors did not show whether their conditional RSPO1 knockout mice impacted Wnt signaling.

We apologize for the confusion caused. We have incorporated more repeats in revised Figure 3C, and our combined dataset suggest there is no significant change in *Axin2* expression after RSPO1 treatments.

Our RSPO1-cKO indeed resulted in decreased Wnt signaling activities in basal cells (see Author response image 2). Control and RSPO1-cKO mice were administered with tamoxifen at 8-week-old and mammary glands were harvested after 4 weeks. Basal cells were isolated for qPCR analysis. We found that the expressions of Wnt target genes, *Axin2* and *Lgr5*, are reduced.

**Author response image 2. respfig2:** 

References:1) Luo, J. et al. Regulation of bone formation and remodeling by G-protein-coupled receptor 48. Development 136, 2747-2756, doi:10.1242/dev.033571 (2009).2) Du, B. et al. Lgr4/Gpr48 negatively regulates TLR2/4-associated pattern recognition and innate immunity by targeting CD14 expression. J Biol Chem 288, 15131-15141, doi:10.1074/jbc.M113.455535 (2013).3) Weng, J. et al. Deletion of G protein-coupled receptor 48 leads to ocular anterior segment dysgenesis (ASD) through down-regulation of Pitx2. Proc Natl Acad Sci U S A 105, 6081-6086, doi:10.1073/pnas.0708257105 (2008).